

# Lucky planets: How circum-binary planets survive the supernova in one of the inner-binary components

F. Fagginger Aeur and S. Portegies Zwart[⋆]

Leiden Observatory, Leiden University, PO Box 9513, 2300 RA, Leiden, The Netherlands

⋆ spz@strw.leidenuniv.nl

## Abstract

A planet hardly ever survives the supernova of the host star in a bound orbit, because mass loss in the supernova and the natal kick imparted to the newly formed compact object cause the planet to be ejected. A planet in orbit around a binary has a considerably higher probability to survive the supernova explosion of one of the inner binary stars. In those cases, the planet most likely remains bound to the companion of the exploding star, whereas the compact object is ejected. We estimate this to happen to $\sim 1/33$ the circum-binary planetary systems. These planetary orbits tend to be highly eccentric ($e \gtrsim 0.9$), and $\sim 20\%$ of these planets have retrograde orbits compared to their former binary. The probability that the planet as well as the binary (now with a compact object) remains bound is about ten times smaller ($\sim 3 \cdot 10^{-3}$). We then expect the Milky way Galaxy to host $\lesssim 10$ x-ray binaries that are still orbited by a planet, and $\lesssim 150$ planets that survived in orbit around the compact object's companion. These numbers should be convolved with the fraction of massive binaries that is orbited by a planet.



# 1 Introduction

Since the discovery of exoplanets around pulsars [1] there has been a debate on their origin. Popular scenarios include in situ formation [2, 3] or the dynamical capture of a planet in a dense stellar system [4]. The possibility of a planet surviving its host star's supernova is often neglected, because a planet in orbit around a single exploding star is not expected to survive the supernova [5]. The combination of mass loss in the supernova explosion [6] and the natal kick imparted to the new compact object [7, 8] mediates the survivability of low-mass x-ray binaries [9–15], but planetary orbits are too fragile to survive this process [16].

The survivability of a planet in orbit around a binary, of which one of the components experiences a supernova is considerably larger than when the planet directly orbits the exploding star. The lower relative mass loss in a binary compared to a single star and the dilution of the velocity kick by dragging along the companion star [17], may cause the planet to survive either in orbit around the binary, or around the original secondary –non-exploding– star (in which case the compact object is ejected from the system).

We calculate the probability that a circum-binary planet survives the first supernova in a massive binary. In a first step, we perform binary population synthesis calculations to determine the orbital phase-space distribution of the pre-supernova binary system. We subsequently add a planet in orbit around the binary and analytically calculate the supernova's effect on the system to determine the planet's survivability and post-supernova orbital parameters.

# 2 Method

We approach the problem on the survivability of circum-binary planets using a combination of techniques. First, we perform a series of population synthesis calculations for binary stars. The binaries that survive until they experience their first supernovae are, in the second step, provided with a circum-binary planet in a stable orbit, after which we resolve the supernova explosion. This second step is calculated analytically for each individual system, and a population study is carried out through Monte Carlo sampling.

## 2.1 Population synthesis calculations of pre-supernova binaries

Binary evolution is a complicated non-linear problem that is not easily performed analytically (see however [14]). Therefore we perform this part of the calculation numerically, using the publicly available and well-tested binary population-synthesis code SeBa [15, 18]. We adopt the version available in the Astrophysics Multipurpose Software Environment (AMUSE, [19]). The code takes the metallicities and masses of the two stars ($m_1$ and $m_2$, at zero age) and together with the orbital period and the eccentricity ($e_i$), the code gives the evolution of these parameters as a function of time. We adopt Solar metallicity throughout this study.

There are quite a number of free parameters in a binary population synthesis code [20]. The most important ones are the treatment of non-conservative mass transfer, the amount of angular momentum per unit mass that is lost by the mass leaving the binary system ($\beta$), and the common-envelope treatment (often summarized in the parameters $\alpha\lambda$ and $\gamma$). We adopt the model parameters as in [12] (their model C, see table 1, $\alpha\lambda = 0.5$, $\beta = 6$ and $\gamma = 1$), which matches with the Galactic x-ray binary population. These parameters are classically identified with $\alpha$, $\beta$, and $\gamma$, but in the next section we use the same letters in a different context.

We determine the probability density function for pre-supernova parameters in phase space through binary population synthesis, and subsequently, apply a kernel-density estimator to smooth these distributions and bootstrap the number of systems. In the next step, we use

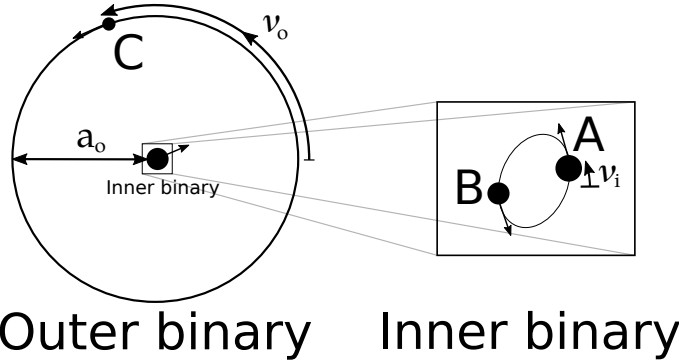

Figure 1: The geometry of the hierarchical triple, with inner binary composed o stars *A* and *B*, and the outer planet *C*. Also indicated are the outer binary semi-major axis $a_o$ and the true anomaly $\nu_i$ and $\nu_o$, for the inner and outer orbit, respectively.

these smoothed distributions to randomly select pre-supernova binaries to which we add a planet and subsequently apply the effect of the supernova.

## 2.2 Analytic considerations of the post-supernova system

The effect of a supernova on a binary system was explored by [6]. Later [11, 15, 21] further studied the binary's survivability through population synthesis, expanding the original formulation for [6] to include elliptical orbits and a wide range of velocity kicks imparted to the newly formed compact object. [22] and [23] subsequently expanded the formalism to multiple systems by considering a hierarchy of nested binaries. We expand on these studies by adopting a planet around a binary system of which one component instantaneously loses mass isotropically and receives a velocity kick.

In figure 1 we sketch the adopted configuration, in which an inner binary (star *A* and star *B*) is orbited by a planet $(C)$[1]. The properties of the inner (outer) binary are labeled with subscript $i$ $(o)$. The star *A* undergoes the supernova. We further adopt the notation from [21], and with that the remnant mass $m'_A$ and the average orbital velocity $v_{\mathrm{orb}}$.

$$\tilde{m} = \frac{m_A + m_B}{m'_A + m_B}, \qquad \tilde{v} = \frac{|\mathbf{v}_{\mathrm{kick}}|}{v_{\mathrm{orb}}}, \qquad \mu_B = \frac{m_B}{m_A + m_B}.$$

For a circular binary we have the expression for the orbital energy after the supernova:

$$E' = -\frac{Gm'_A m_B}{2a}\Delta,$$

$$\Delta = \frac{2a}{|\mathbf{r} - \delta\mathbf{r}|} - \tilde{m}(1 + \tilde{v}^2 - 2\tilde{v}\cos(\theta)\cos(\phi)).$$

Here $\theta, \phi$ give the kick direction (we define $\theta = \phi = 0$ to be a kick anti-parallel to orbital motion, and therefore have a minus sign difference with [21]), $a$ is the semi-major axis of the binary before the supernova, and $\mathbf{r}$ the relative position vector between the binary components. The shift $\delta\mathbf{r}$ allows for an instantaneous displacement of the center of mass of the inner binary. This displacement of the inner binary results from supernova shell when it passes the companion star (see [22]). A bound orbit requires $E' < 0$ and therefore $\Delta > 0$. This inequality

---

[1]Following [15], we write systems by assing the first three letters of the alphabet to the two stars and the planet. Parenthesis indicate a bound system, and multiple braces a hierachy. The initial triple, for example, then is written as $((A, B), C)$, and a possible binary between the seconday star and the planet with an unbound primary is written as $A, (B, C)$.

gives the physical requirements on the mass loss and velocity kick. Combining this with the method of [22] we can derive a similar formula for the outer binary in the hierarchical triple. We assume that both binaries are initially circular and co-planar (see figure 1). We select the inner binary's true anomaly $\nu_i = 0$ by an appropriate rotation of our coordinate system and assume $m_C \ll m_A, m_B$. We define a $\Delta_i, \Delta_o$ for both binaries. The supernova in the inner binary leads to an 'effective' supernova in the outer binary through an instantaneous change in the mass, position, and velocity of the inner binary center of mass:

$$\tilde{m}_o = \frac{(m_A + m_B) + m_C}{(m'_A + m_B) + m_C} \approx \frac{m_A + m_B}{m'_A + m_B} = \tilde{m}_i \equiv \tilde{m},$$

$$\delta \mathbf{r}_{\text{COM}} = -\mu_B(\tilde{m} - 1)a_i \hat{\mathbf{r}}_i,$$

$$\delta \dot{\mathbf{r}}_{\text{COM}} = v_{\text{orb},i}[-\mu_B(\tilde{m} - 1)\hat{\mathbf{r}}_i + (1 - \tilde{m}\mu_B)\tilde{v}_i \hat{\mathbf{v}}_{\text{kick}}].$$

We define auxiliary variables:

$$l = a_o/a_i,$$

$$K_1 = \mu_B(\tilde{m} - 1) = \mu_B(m_A - m'_A)/(m'_A + m_B),$$

$$K_2 = 1 - \tilde{m}\mu_B = m'_A/(m'_A + m_B),$$

$$\alpha = \left(1 + (K_1 l^{-1})^2 + 2K_1 l^{-1} \cos(\nu_o)\right)^{-1/2},$$

$$\beta = \left(1 + K_1^2 l + 2K_1 \sqrt{l} \cos(\nu_o)\right)^{1/2},$$

$$\gamma = K_2 \sqrt{l},$$

$$\phi_0 = \arctan\left(\frac{\sin(\nu_o)}{\cos(\nu_o) + K_1 \sqrt{l}}\right) + \frac{\pi}{2}\left(1 - \text{sgn}\left(\cos(\nu_o) + K_1 \sqrt{l}\right)\right).$$

Letting $\tilde{v} = \tilde{v}_i$, this leads to expressions for $\Delta_{i,o}$:

$$\Delta_i = 2 - \tilde{m}[1 + \tilde{v}^2 - 2\tilde{v}\cos(\theta)\cos(\phi)],$$

$$\Delta_o = 2\alpha - \tilde{m}[\beta^2 + (\gamma\tilde{v})^2 + 2\beta\gamma\tilde{v}\cos(\theta)\cos(\phi - \phi_0)].$$

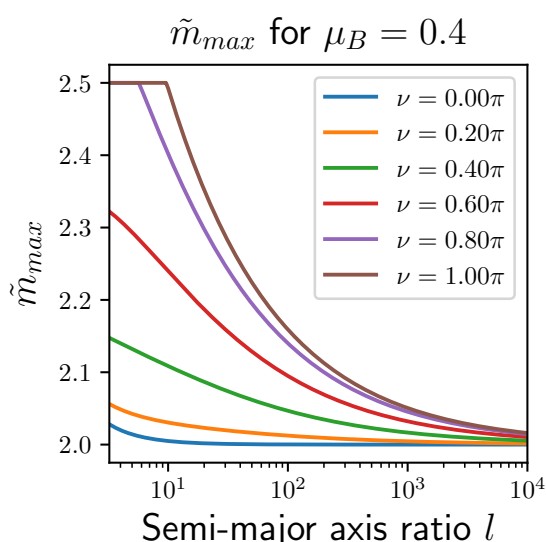

Figure 2: Maximum relative mass loss $\tilde{m}_{\text{max}} = (m_A + m_B)/(m'_{A,\text{min}} + m_B)$ as a function of semi-major axis ratio $l = a_o/a_i$ for various outer binary true anomalies.

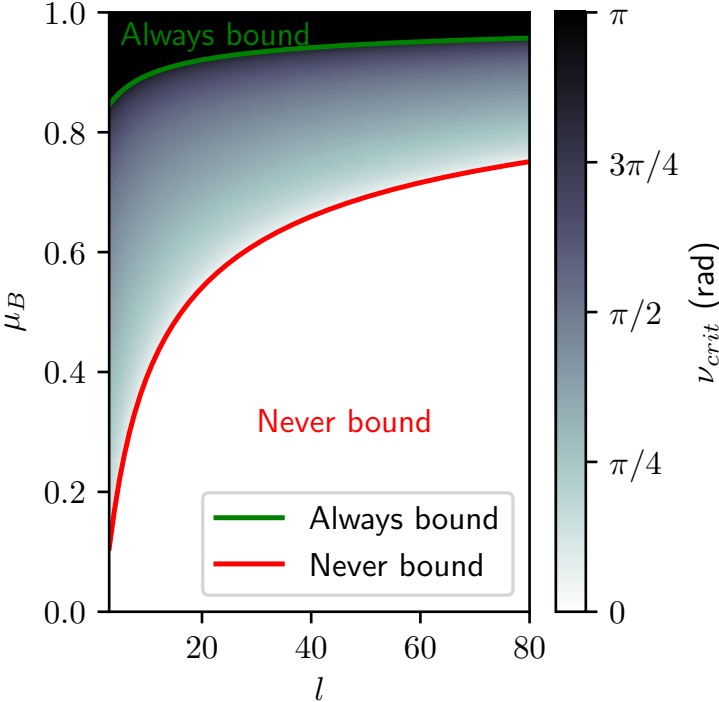

Figure 3: Boundaries in $\mu_B, l, \nu_o$ phase space for which the $(B, C)$ subsystem remains bound when the remnant star $A$ is ejected from the system. The system $(B, C)$ remains bound if $\nu_o \in (\pi - \nu_{\text{crit}}, \pi + \nu_{\text{crit}})$.

Note that each of the terms in $\Delta_o$ are re-scaled versions of similar terms in $\Delta_i$. Each of the factors $\alpha, \beta, \gamma$ gives the effect of the supernova in the inner binary on the outer binary. The effect of kick direction compared to the planet's true anomaly on the outer binary survivability is given by $\phi_0$. We can obtain limits on the magnitude of the kick and its direction by following [21]. A new constraint for the triple system is the existence of a maximum $\tilde{m}$. From the inequalities $\Delta_i > 0, \Delta_o > 0$ we get requirements on the kick magnitude. Specifically, $\tilde{\nu}$ must be in the intervals for the inner and outer orbits:

$$\left[1 - \sqrt{\frac{2}{\tilde{m}}}, 1 + \sqrt{\frac{2}{\tilde{m}}}\right] \quad \text{and} \quad \left[\frac{\beta}{\gamma} - \sqrt{\frac{2\alpha}{\tilde{m}\gamma^2}}, \frac{\beta}{\gamma} + \sqrt{\frac{2\alpha}{\tilde{m}\gamma^2}}\right].$$

In the limit $m'_A \to 0$ the value of $\gamma^{-1}$ diverges, and the lower bound of the second interval may exceed the upper bound of the first interval (depending on its sign). In such a case, the system cannot remain bound, leading to a maximum allowed mass loss $\tilde{m}_{\text{max}}$. This value is limited by the constraint $m'_A \geq 0$, which sets $\tilde{m} \leq \tilde{m}_{\text{max}} \leq \mu_B^{-1}$. This is plotted in figure 2.

The survivability of the $(B, C)$ as a bound subsystem with $A$ ejected can be investigated by letting $m'_A \to 0$ in $\Delta_o$. Examining the limits $\Delta_{(B,C)} = 0$ gives the boundaries in $\mu_B, l, \nu_o$ phase space where $(B, C)$ remains bound. This is plotted in figure 3. For each pair $\mu_B, l$ we maximize $\Delta_{(B,C)}$ by picking $\nu_o = \pi$. Using that $\cos(x)$ is an even function and that we maximize $\Delta_{(B,C)}$ by setting $\nu_o = \pi$, we can define a $\nu_{\text{crit}}(\mu_B, l)$ so that $\Delta_{(B,C)} > 0$ if $\nu_o \in (\pi - \nu_{\text{crit}}, \pi + \nu_{\text{crit}})$. For a uniformly random value of $\nu_o$ at the time of the supernova the probability of the planet remaining bound is $\nu_{\text{crit}}/\pi$.

Table 1: Initial parameter distributions and value ranges for the inner eccentricity $e_i$, mass $m_1$, mass ratio $m_2/m_1$, and orbital period $P$. the primary mass-function and eccentricity limits are adopted from [24] and [25], respectively.

| Parameter | Distribution shape | Value range |
|---|---|---|
| $m_1[\mathrm{M_\odot}]$ | $(m_1[\mathrm{M_\odot}])^{-2.3}$ | $[10, 100]$ |
| $m_2/m_1$ | $(m_2/m_1)^{0.1}$ | $[6 \cdot 10^{-3}, 1]$ |
| $\log_{10}(P[\mathrm{d}])$ | $\log_{10}(P[\mathrm{d}])^{0.2}$ | $[0, 4]$ |
| $e_i$ | $e_i^{-0.6}$ | $[10^{-4}, 0.9]$ |

## 3 Results

Having set out the framework for determining the pre-supernova binary properties, and describing the effect of the supernova on the triple, we now combine both to study the survivability of a circum-binary planet.

### 3.1 Specific choice of initial conditions

We initialize zero-age main-sequence binaries and evolve them using population synthesis (see section 2.1). In table 1 we give the initial conditions.

Binaries that merge or become unbound before the supernova occurs are discarded. We subsequently bootstrap the number of systems by training an `sklearn` kernel density estimator on $\ln(m_A)$, $\ln(m_B)$, $\ln(a_i)$, $\ln(e_i/(1-e_i))$, $\ln(a_{i,\max})$, and the logarithm of the remnant mass. This ensures that the values drawn from this distribution are positive and $e_i$ is in the range $(0, 1)$. Here $a_{i,\max}$ is the largest semi-major axis encountered during the evolution of the inner binary and $e_i$ the inner binary's eccentricity.

The mass of a black hole is determined by `SeBa`, but for the neutron-star mass we adopt a Gaussian distribution with mean $1.325\,M_\odot$ and standard deviation of $0.1125\,M_\odot$ independent of the progenitor mass (figure 2 in [26]). The other parameters for the supernova kick are drawn randomly from the distribution functions presented in table 2.

The inner binary is generally circularized due to mass transfer before the supernova occurs. In some wide binaries (those with orbital periods $\gtrsim 20\,\mathrm{yr}$), however, this may not be the case. These wide and eccentric binaries tend to be very fragile for the supernova, in particular the planet in an even wider and stable orbit around such binaries are susceptible to being ionized as a result of the supernova explosion. In some cases, however, the combination of mass loss and the natal supernova kick may cause the planet to remain bound. To accommodate this possibility, we relax the assumption of a circular inner binary just before the supernova. In our Monte Carlo sampling, we randomly select the true anomaly of the inner binary to determine the effect of the supernova. A uniformly distributed true anomaly introduces a bias towards smaller separations and consequently higher orbital velocities in comparison with adopting a uniform distribution in the mean anomaly. We correct for this effect by determining, for each binary, the probability distribution of mean anomalies by weighting the simulation results, and scale the supernova survival probability accordingly.

### 3.2 Evolution of the inner binary

After having determined the initial parameter space, we evolve a total of $4.8 \times 10^4$ zero age binaries up to the moment of the first supernova.

A fraction of 0.14 experienced a supernova resulting in a black hole, 0.08 produced a neutron star in a supernova explosion, 0.03 left no remnant after the supernova; the rest

Table 2: Distribution parameters for the hierarchical triple and velocity kick. Here $e$ refers to eccentricity, $i$ to inclination, and $\omega$ to argument of periapsis. The parameter $a_{i,\max}$ is the maximum inner binary semi-major axis found during the SeBa evolution. The lower and upper limits to the semi-major axis are from [27] and [28], respectively.

| Parameter | Distribution shape | Value range /parameters |
|---|---|---|
| $a_o/a_{i,\max}$ | $(a_o/a_{i,\max})^{-1}$ | $[3.23, 1000]$ |
| $\|\mathbf{v}_{\text{kick}}\|[\text{km/s}]$ | Maxwellian | $\sigma = 265$ |
| Kick direction | Uniform | Sphere |
| $m_C[M_\odot]$ | Uniform | $[10^{-5}, 5\cdot 10^{-3}]$ |
| $e_o$ | - | 0 |
| $i_o$ | - | 0 |
| $i_i$ | - | 0 |
| $\nu_o$ | Uniform | $[0, 2\pi]$ |
| $\nu_i$ | Uniform | $[0, 2\pi]$ |
| $\omega_i$ | Uniform | $[0, 2\pi]$ |

either resulted in a merger, an unbound system or did not experience a supernova. We do not discuss the evolution of these systems here, but adopt the eventual distributions of the orbital parameters of the surviving binaries to further explore the possibility that a circum-binary planet survives the supernova.

Figure 4 shows the parameter distributions for the simulations leaving a remnant. Most of the supernova progenitors have a mass $m_A < 10\,M_\odot$, while there are secondary masses as high as $80\,M_\odot$. Most of the supernovæ are stripped core-collapse of type Ibc that naturally result from the mass transfer during the system evolution [29]. Most inner binaries have semi-major axes $< 1$ au.

## 3.3 Survivability of the planet after the supernova

In order to study the probability that a planet survives in orbit throughout the supernova, we add a planet, with a mass chosen uniformly between $10^{-5}\,M_\odot$ and $5 \times 10^{-3}\,M_\odot$ and orbital separation between $a_o = 3.23\,a_{i,\max}$ to $1000\,a_{i,\max}$ (flat in $\log(a_o)$). Here $a_{i,\max}$ is the largest semi-major axis the inner binary reaches during its pre-supernova evolution. The inner limit is chosen to assure that the planet would remain stable throughout the evolution of the inner binary ( [30, 31], and see [28] for a more empirical characterization). Mass lost from non-conservative mass transfer in the inner binary or by the wind of one of the components will have driven the planet further out [32], but we ignore that here. The planets are chosen to have prograde circular orbits in the plane of the pre-supernova binary system.

The supernova is simulated applying an instantaneous mass loss and change in the velocity vector of star $A$. This leads to a sudden change in the center of mass and velocity of the inner binary, which has the effective of a (diluted) supernova in the outer orbit, as discussed in [22].

Neutron stars and black holes acquire a velocity kick upon their formation in a supernova. This kick's magnitude and direction are crucial for the survivability of the binary system and important for the orbital parameters when the binary remains bound. We adopt the distribution of pulsar kicks from [27], which is described by a Maxwellian distribution with a dispersion of $\sigma = 265$ km/s. This distribution appears consistent with population statistics of x-ray binaries in the Milky Way [12]. Black holes are expected to receive a lower kick velocity, which we address by applying a momentum-kick.

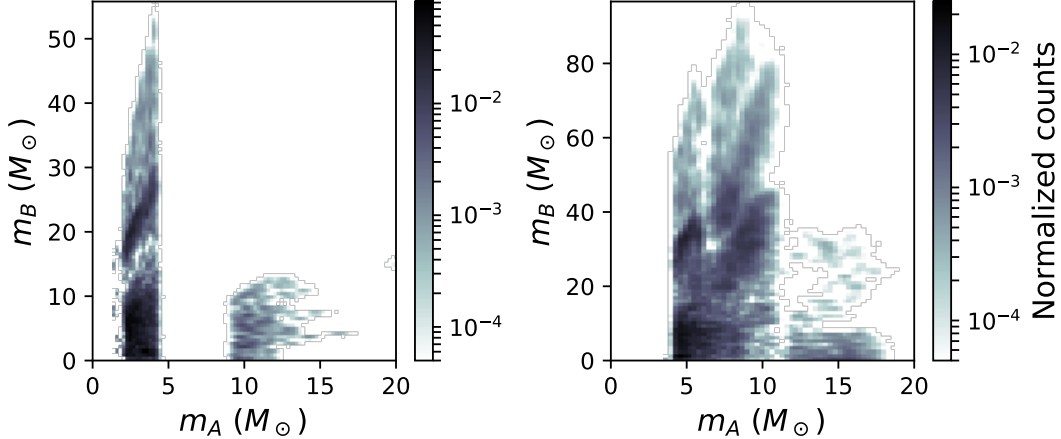

(a) Pre-supernova masses for binaries with a neutron star.

(b) Pre-supernova masses for binaries with a black hole.

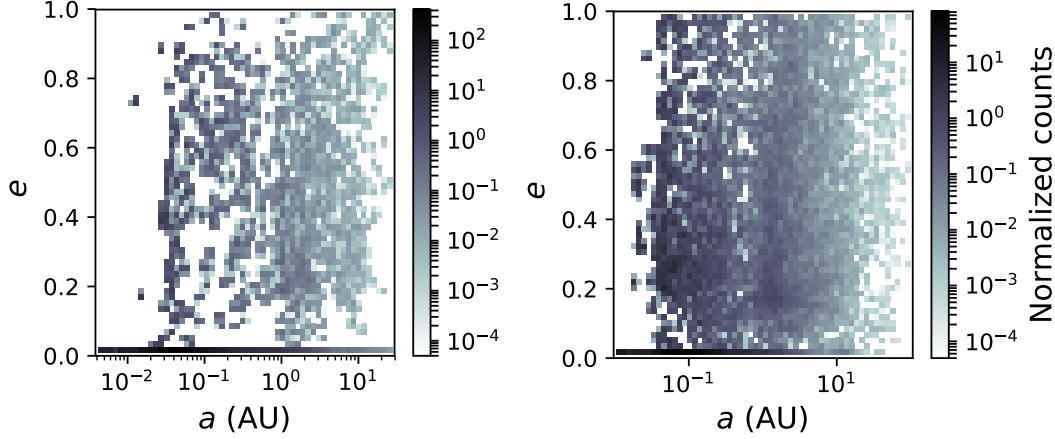

(c) Semi-major axis versus eccentricity for neutron-star binaries directly after the supernova

(d) Semi-major axis versus eccentricity for black-hole binaries directly after the supernov.

Figure 4: Distribution of the parameters for the inner binaries just before and after the supernova for both neutron stars (left) and black holes (right).

The supernova in a triple then results in one of the following:

- The entire triple becomes unbound: $A, B, C$,

- The triple becomes dynamically unstable,

- The inner binary remains bound, but the planet escapes: $(A, B), C$,

- $A$ and $C$ remain bound, but $B$ escapes: $(A, C), B$,

- $B$ and $C$ remain bound, but $A$ escapes: $A, (B, C)$,

- The entire triple remains bound: $((A, B), C)$,

We look at every possible combination of $A, B$ and $C$ and determine the distribution of the corresponding orbital elements. We recognize two distinct dynamically unstable configurations: 1) the entire triple remains bound but violates the stability criterion [28], and 2) the is no

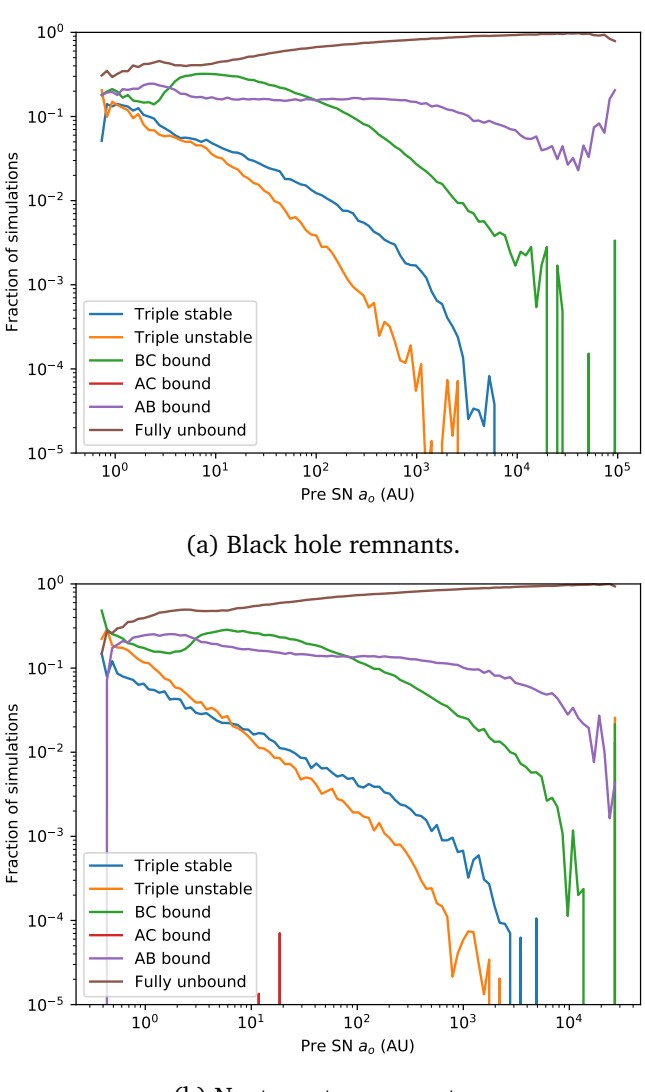

(a) Black hole remnants.

(b) Neutron star remnants.

Figure 5: Fraction of simulations per possible outcome of the supernova as a function of the outer binary semi-major axis $a_o$ before the supernova. The fraction of simulations in these figures is calculated as the ratio of two histograms of $a_o$ with logarithmic bins, comparing the number of simulation with a specific outcome to the total number of simulations in each bin. The 'triple unstable' category refers to systems which remain fully bound but for which the outer planet orbit is unstable [30].

clear hiearcy in the surviving triple. In the latter case we often find that The planet is bound to both stars, but the two stars are not bound in a binary.

Using the earlier mentioned binary population synthesis results, we generate $4 \times 10^6$ pre-supernova systems with a neutron star and $4 \times 10^6$ with a black hole, each of which with a circum-binary planet.

In figure 5, we show, as a function of $a_o$, the fraction of simulations that experience a supernova (see 2.1). The increasing noise for $a_o > 10^3$ au is due to the smaller number of simulations in that region of parameter space (note that $a_0$ was initially distributed randomly on a log scale). Another interesting feature is that for $a_o \lesssim 10^2$ au the probability of $(B, C)$ to remain bound is larger than for $(A, B)$ to stay bound. The majority of triples that survive the supernova are dynamically stable, which is a direct consequence of the adapted stable conditions before the supernova.

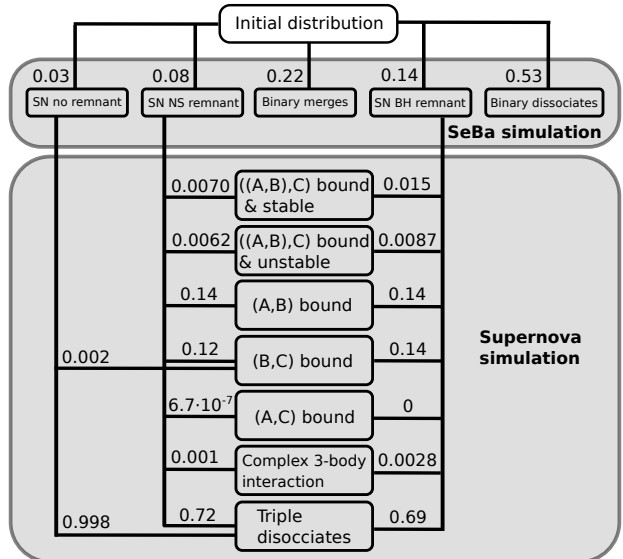

Figure 6: Branching ratios for each possible outcome for the triple system, averaged over all input parameters. The upper branches give the results of the binary population synthesis, and the lower branch gives the effect of the supernova on the resulting systems.

In only 3 of the $8\times10^6$ simulations, the planet remains bound to the exploding star, leading to a binary composed of $(A, C)$. For this to happen, the remnant must receive a kick in a narrow cone and with the velocity similar to the planets orbital speed but away from the companion star so that it can pick-up the planet on its escaping trajectory. Alternatively, the compact object escapes, leaving it companion star $B$ bound to the planet. Both processes are rather improbable, as expected, which is consistent with the small probability in our simulations.

In figure 6 we present the branching ratios of all occurrences in our simulations. Following the appropriate branches, the total probability for the triple to survive is $2.7 \cdot 10^{-3}$, and the probability that $(B, C)$ remains bound is $3 \cdot 10^{-2}$. The currently detected number of x-ray binaries is around 300 [33, 34], with an estimates for the Galactic population of $\sim 1300$ [35] to $10^4$ [36]. With these last two estimates, the expected number of x-ray binaries that after the supernova that still host a circum-binary planet is $\lesssim 10$: the Galaxy may host a few x-ray binaries with a circum-binary planet that survived its host's supernova. There is a comparable probability for the entire triple to remain bound but with an unstable orbit for the planet. In the latter case, the planet may either collide with one of the stars or be ejected to become a rogue planet.

In figure 7 we present the distribution of the semi-major axis and eccentricity of the surviving planet's orbit. The planet then either continues to orbit the inner binary, or it orbits the original secondary star. Mass loss in the inner binary generally induces a considerable eccentricity in the final orbit. Surviving planets therefore typically have highly eccentric orbits, with a wide range of semi-major axes.

There is an absence of stable triples with semi-major axes larger than $10^3$ au and low eccentricities, because the number of planets with a pre-supernova semi-major axis $> 10^3$ au is low, and the post-supernova periapsis of the planet can not exceed the pre-supernova apoapsis. No planets are found with tight $\ll 1$ au orbits because these systems tend to be dynamically unstable, and wider, up to $\sim 10$ au orbits tend to have high eccentricities. Most of the orbits in the $(B, C)$ systems that remain bound have $e_{(B,C)} \uparrow 1$ due to the extreme mass loss in the triple. We predict systems that formed this way to have highly eccentric planetary orbits, with no strong constraints on the semi-major axis.

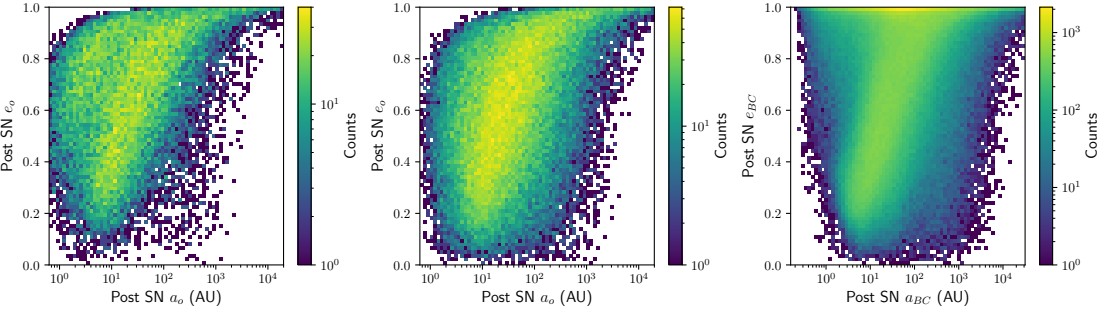

(a) Black hole remnants.  (b) Neutron star remnants.  (c) Case for which the exploding star is ejected.

Figure 7: Distribution for semi-major axis and eccentricity for the orbiting planet that survive the supernova of one of the binary components. The panels a) and b) give the orbital elements for the circum-binary planet with a black hole and a neutron star, respectively. Panel c) gives the orbital parameters for the planet in orbit around star $B$.

In figure 8 we present the distribution of relative inclinations for the planetary orbits with respect to the post-supernova binaries. In this case, we used the results for the neutron stars, but the black hole distribution is similar. The fraction of retrograde orbits for circum-binary planets with a neutron star is 0.277 compared to 0.148 for binaries with a black hole. A fraction of 0.206 of the planets around a companion of the exploding star in $(B, C)$ has a retrograde orbit.

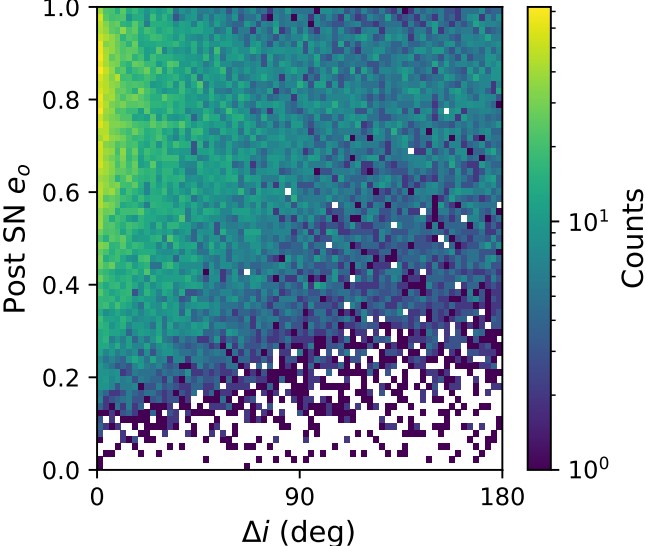

Figure 8: Relative inclination of the planet orbit with respect to the post-supernova binary, presented in the case where the supernova resulted in a neutron star. $\Delta i = 0$ indicates that the planet still orbits in the plane of the post-supernova binary system. The fraction of retrograde orbits is between 0.148 (for binaries with a black hole) and 0.277 (for binaries with a neutron star).

# 4 Observational implications

When both stars and the planet remain bound, the inner binary hosting the compact object continues to evolve into an x-ray binary. In that case, the circum-binary planet may be observed in the x-ray binary phase or around a binary millisecond pulsar once the neutron star has been spun-up. In principle, it is even possible that the planet survives a second supernova explosion, in which case the planet remains bound to the inner compact-object binary. This requires the triple to survive and remain stable through both supernovæ. The probability of this to happen seems small ( $\lesssim (0.14 \times 0.015)^2 \sim 4.4 \cdot 10^{-6}$, for a black hole binary and more than an order of magnitude smaller for neutron star binaries, ( $\lesssim (0.08 \times 0.00070)^2 \sim 3.1 \cdot 10^{-7}$, see figure 6). When the compact object is ejected, the planet may still survive in a relatively wide and eccentric orbit around the original secondary star. Such systems may be recognizable for their curious planetary orbit and the mass-transfer affected stellar host.

The orbits of the surviving planets are wide, with semi-major axes ranging from $\sim 1$ au to well over $10^3$ au, and over the entire range of eccentricities, but skewed to $e \gtrsim 0.2$ orbits. The majority of systems in which the planet remains bound to the companion of the exploding star tends to have very highly eccentricities $e \gtrsim 0.9$ orbits (but skewed to $e \sim 1$). Such a system may lead to a merger between the planet and its newly acquired companion star within a few years after the supernova. For the surviving systems, the discovery of a massive star with a planet in a wide (10 to $10^3$ au) and eccentric ( $\gtrsim 0.9$) orbit may be a signature of a supernova survival. These planets will have experienced a nearby supernova, possibly obliterating their atmosphere, or at the least enriching it with a healthy radioactive mix of heavy decay-product of the supernova blastwave.

A considerable fraction of the pre-supernova systems have eccentricities $e > 0.1$ (0.24 of systems with a neutron star and 0.32 of systems with a black hole). A uniformly random true anomaly, as we adopted in our analysis, is biased to lower separations and higher orbital velocities compared to a mean anomaly selected randomly from a uniform distribution for eccentric systems. This impacts the survivability of the system, and introduces a bias in our results. We correct for this by calculating how much the mean anomaly of a specific system is overcounted (compared to a uniformly random mean anomaly), and scaling the effect of that system on our results by the inverse of this.

# 5 Conclusion

We simulated a population of massive zero-age binaries up to the moment of the first supernova. The surviving binaries were equipped with a circum-binary planet to determine the probability distributions of the planets' orbital parameters of the surviving binaries. We did this by analytically investigating limiting factors on the planet's survival and through Monte Carlo sampling. The analytic expressions for the amount of mass lost to assure that the planet remains bound and the probability of remaining bound are presented in figures 2 and 3, respectively. The resulting numerically calculated survivability for the planet is presented in figure 5.

From the total population of massive Galactic binaries that evolve into x-ray binaries, we predict a fraction of $3 \cdot 10^{-3}$ to keep its circum-binary planet. This fraction should be perceived as an upper limit, because we assumed 100% triplicity. Interestingly enough, the probability for the planet to remain bound to the exploding star's companion is 11 (0.03/0.0027, see the table in figure 6) times higher, or $3 \cdot 10^{-2}$. These systems, however, are probably harder to identify as post-supernova planetary systems except maybe for the curious orbit of the single planet. More than 20% of these planets have retrograde orbits compared the their pre-suprenova orbit.

This could potentially be observed in mass transfer in the pre-supernova epoch has synchronized the secondary's rotation.

We conclude that $\lesssim 10$ x-ray binaries in the Galaxy may still harbor a circum-binary planet, and at most $\sim 150$ massive stars may be orbited by a planet in a wide ($\gtrsim 10$ au) and highly eccentric ($\gtrsim 0.9$) orbit. Note, however, that we assumed that every binary is orbited by a planet, which seems unlikely (see [37]).

The survivability of the circum-binary planets is mediated by the large proportion of stripped (post-mass transfer) supernovae in our simulations. Such a type Ib/c supernova results in a relatively small mass loss, which helps to keep the planet bound. Together with the probability of a low kick velocity, mediates in the survivability of planets around post-supernova systems. The adopted distribution functions for the kick velocity are important for this study because of the planet's survival. There are not many constraints to the low-velocity tail of the supernova kick distribution [44]. In a future study, it may be worth exploring this part of parameter space more exhaustively. Finally, our conclusions should be checked against observations.

## Energy consumption of this calculation

The population synthesis calculations for $4.8 \times 10^4$ zero-age binaries took about 3 seconds per binary on a single Xeon E-2176M core ($\sim 12$ Watt) resulting in about 41 hours of CPU time or 0.5 kWh. Building the database and repeating the calculations because of earlier errors or tuning the selected initial conditions we should multiply the runtime by a factor of 2 or 3. With 0.649 kWh/kg (Dutch norm for gray electricity) results in $\sim 2$ kg CO2, which is comparable to a daily commute.

## Software used for this study

In this work we used the following packages: `python` [38,39], `SeBa` [15,18], `AMUSE` [19], `numpy` [40], `scipy` [41], `sklearn` [42], `matplotlib` [43], and `sqlite3`.

## Acknowledgments

We thank the anonymous referee for spotting an error in our earlier version of the manuscript, and for useful comments that helped us improving the manuscript. This work was performed using resources provided by the Academic Leiden Interdisciplinary Cluster Environment (ALICE).

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
