# Peer review of "Lucky planets: how circum-binary planets survive the supernova in one of the inner-binary components"

_SciPost Astronomy, doi:SciPost Astro. 2, 002 (2022)_

## Round 1 · Referee Report · Anonymous (Referee 1) · 2021-3-11

Strengths

1- In this work, the authors tackle the novel idea of (circumbinary) planet survivability beyond the main sequence. 2-Heuristic or semi-quantitative arguments about "second generation" planets or "Phoenix" planets around stellar remnants have been proposed on-and-off for the past ~30 years. However, detailed Monte Carlo experiments are less numerous, and thus such degree of novelty makes this paper worthy of publication

Weaknesses

1- The paper is verbose and its main message is muddled with extraneous arguments and caveats. 2- It reads to me like a null result, but without directly presenting itself as such. 3- Some figures seem unnecessary.

Report

Dear Scientific Editor:

I have carefully read the Scipost manuscript 2101.08033v1 entitled "Lucky planets: how circumbinary planets survive the supernova in one of the inner-binary components". In this work, the authors tackle the novel idea of (circumbinary) planet survivability beyond the main sequence. In particular, the authors study the fates of planets around massive binaries that undergo supernova explosion. Via simple analytic arguments and Monte Carlo simulations, the authors compute the fraction of systems that could harbor a planet after the supernova and discuss the potential observational signposts of such processes.

MAIN IMPRESSION: While certainly publishable, this paper could benefit from further streamlining its message: ideally, it should be direct and concise, and not weakened by its own counterpoints/caveats, or diluted by open ended speculations and pessimistic takes on how observable the theoretical predictions are.

GENERAL COMMENTS: Heuristic or semi-quantitative arguments about "second generation" planets or "Phoenix" planets around stellar remnants have been proposed on-and-off for the past ~30 years. However, detailed Monte Carlo experiments are less numerous, and thus such degree of novelty makes this paper worthy of publication.

However, the tone and purpose of this paper is ambiguous. The Introduction opens with a discussion on the pulsar planets, making the reader think that this paper is about mechanisms of producing such systems. If "pulsar planet production from pre-existing circumbinary planets" is indeed the hypothesis being tested here, then this paper is reporting a null result. And null results are fine. But neither in the abstract, the conclusion, nor anywhere in the narrative of the paper is categorically stated that the inferred hypothesis (perhaps I misunderstood the intended one) has been proven inadequate.

Thus, for good writing's sake, the authors need to decide if this is a null result (which is welcome) or something else, but the disconnect between the Introduction and the rest of the text is too striking not to be addressed. It appears as if the Introduction was written before the results came in.

Just to reiterate: this is careful, quantitative work, and is deserving of a more effective (and ideally shorter) delivery than is currently being given. Moreover, the readership of this journal (and other researchers working on related topics) will benefit from a straightforward description of the results, even if they seem less exciting that originally expected. In summary, directly saying "pulsar planets CANNOT form this way", is an informative result.

OTHER MAJOR COMMENTS: - Section 2.2, Second paragraph: Much like in Brandt & Podsiadlowski (1994), it would be useful to the reader to clarify that the semi-major axis a in the definition of E' is the old semi-major axis and distinct from the post-explosion semi-major axis a'.

  • Section 2.2, Second paragraph: At the time of mass loss, why is the true anomaly nu_i chosen to be zero at all times? Should not the mean anomaly (i.e., time) be sampled randomly from 0 to 2pi? For eccentric binaries, choosing nu_0=0 is certainly not "without loss of generality" and might affect some of the results. Case in point: in Section 3.3, Sixth paragraph, the authors state that for the A,C pair to remain bound,"the remnant must receive a kick in the right direction and with the proper magnitude to either pass close-by the planet C, [...]". If a kick "in the right direction" is needed, how can this scenario not depend on the choice of nu_i at the time of mass loss and velocity kick?

-Section 3.3, First paragraph Care to comment on how the Mardling+Aarset stability criterion compares to the Holman+Wiegert stability criterion (https://ui.adsabs.harvard.edu/abs/1999AJ....117..621H/abstract) which was especially derived for circumbinary systems?

  • Section 3.3, Second paragraph: The bullet point list of this section is very helpful to visualize the potential outcomes the authors are studying. However, the second category "dynamically unstable" could use further clarification. According to the caption of figure 5 (the explanation should be in the main text as well), these are "[...]systems which remain fully bound but become dynamically unstable." If the Mardling & Aarseth triple stability criterion was used to flag these systems, then that citation should be made explicit when this category is introduced (the paper is cited elsewhere in the manuscript).

Also, I would suggest careful wording here. If the post-explosion triple has total negative energy (i.e., remains bound on energetic grounds), and then becomes dynamically unstable with one object being kick out to infinity, there must be a left-over binary with negative energy. So, one cannot go from bound triple to fully dissolved triple from dynamical interactions alone (I am certain the authors know this basic fact, but the text is a bit ambiguous).

Finally, "and undergoes a three-body interaction" seems an unnecessary addition to "becomes dynamically unstable", especially since all triples, in a strict sense, are undergoing three-body interactions.

  • Section 3.3, Fifth paragraph: The sentence: "[...] attribute the decrease in the fraction of (A,B) systems that stay bound for ao > 10^3 au to the systematically larger measure of ai in this region[...]" is difficult to parse. But it also difficult to see why it is worth pointing out why the purple curve decreases slightly for a_o > 1e3...

  • Section 3.3, Eighth paragraph: The authors state that the "probability for stable triples quickly drops for ao > 1e3 au [...]". Is this independent of the initial distribution of planetary semi-major axes? It is clear that distant planets, being weakly bound to begin with, should be more susceptible to being unbound when the central object loses mass. But what is not so clear is whether this 1000 au boundary scales with the initial outer tail of the planet semi-major axis distribution.

  • Section 3.3, Ninth paragraph: How are the authors defining a negative inclination? Relative inclinations are formally defined as arccos(n_i.n_o) where n_i, n_o are the normal vectors of the inner (binary) and outer (planet) orbits, respectively. Thus, relative inclination is always a positive quantity (0-90 is prograde and 90-180 is retrograde). This finding and the discussion around it are very confusing.

-Section 4: This overall Section reads more like a Discussion than a Conclusion. What are the actual findings of this work? Paragraphs 1,2,3 contain raw findings, but they could be significantly condensed. Paragraph 4 and 5 are definitely Discussion material. Paragraph 6 is a conclusion. Paragraph 7 is mostly meandering caveats than bring down the momentum of the paper and end on a slightly negative tone (caveats are good, but ending with caveats is just weird).

-Section 4, Fourth paragraph: The authors skim over a very intriguing possibility (in my humble opinion): a second SN explosion in the triples that survived. In this case, the BC scenario could very well repeat itself and leave us with a pulsar planet. Even more intriguingly, the authors suggest that "the planet sticks to the inner compact-object binary until they merge due to the emission of gravitational waves". This scenario would circle back to some of my objections with the Introduction, i.e., the formation of pulsar planets. Alas, the authors conclude that the probability of this happening is small. Yet, I could not find where the numbers they use came from (except for 0.14).

MINOR COMMENTS: - Section 1, First paragraph: If the authors do keep their initial discussion on pulsar planets, I recommend reading/referencing a fascinating early take on this systems by Phinney & Hansen 1993 (https://ui.adsabs.harvard.edu/abs/1993ASPC...36..371P/abstract) in which the survivabilty of planets beyond the main sequence is discussed.

  • Section 1, Second paragraph: I am well aware that the term "ionize a binary" is widely used in the binary evolution community (despite my own preference for simply "unbind"). However, the expression "ionizes the planet" can be easily misinterpreted. Please change to "ionizes the planetary orbit" and make sure that either "ionize" or "unbind" is consistently used throughout the paper for the sake of clarity: e.g., Figure 5 refers to "unbound" binaries. There is even a "dissociate" in Section 3.1

  • Section 3.1 Second paragraph. What is the real purpose of training a smooth kernel on an empirical distribution? Bootstrap the number of systems? Benefit from the sample of a continuous (and differentiable) function? The benefit of this technique over simply resampling from an empirical histogram should be clearly stated, otherwise it seems like an unnecessary over-sophistication.

TYPOS: The manuscript contains several typos and a few punctuation errors. Here are a few: - Section 1, second paragraph: might works -> might work - Section 1, third paragraph: larger then -> larger than - Section 3.2, first paragraph: zero binaries-> zero eccentricity binaries? - Section 3.3, last paragraph: neutral star -> neutron star

I hope that the authors find these comments constructive

Best regards,

Requested changes

1- Streamline the paper. At the very least reword the Introduction and shorten /strengthen the Conclusions (or move part of the Conclusions to a Discussion section) 2 - Explain/fix the issue of negative inclinations 3- Justify/fix the choice of true anomaly nu_i=0 in their calculations 4 - Explain the category "unstable triple" better

  • validity: good
  • significance: ok
  • originality: good
  • clarity: ok
  • formatting: acceptable
  • grammar: good

Author:  Simon Portegies Zwart  on 2021-06-14  [id 1505]

(in reply to Report 1 on 2021-03-11)
Category:
answer to question

It is a pleasure to respond to the referee report on our manuscript "Lucky planets: how circum-binary planets survive the supernova in one of the inner-binary components". about the survivability of planets in orbit around a binary in which one star experiences a supernova.

Here we respond to the referee's comments.

MAIN IMPRESSION: While certainly publishable, this paper could benefit from further streamlining its message: ideally, it should be direct and concise, and not weakened by its own counterpoints/caveats, or diluted by open ended speculations and pessimistic takes on how observable the theoretical predictions are.

We streamelined the paper, according to the referee's recommendation.

GENERAL COMMENTS: Heuristic or semi-quantitative arguments about "second generation" planets or "Phoenix" planets around stellar remnants have been proposed on-and-off for the past ~30 years. However, detailed Monte Carlo experiments are less numerous, and thus such degree of novelty makes this paper worthy of publication. However, the tone and purpose of this paper is ambiguous. The Introduction opens with a discussion on the pulsar planets, making the reader think that this paper is about mechanisms of producing such systems. If "pulsar planet production from pre-existing circumbinary planets" is indeed the hypothesis being tested here, then this paper is reporting a null result. And null results are fine. But neither in the abstract, the conclusion, nor anywhere in the narrative of the paper is categorically stated that the inferred hypothesis (perhaps I misunderstood the intended one) has been proven inadequate.

Thus, for good writing's sake, the authors need to decide if this is a null result (which is welcome) or something else, but the disconnect between the Introduction and the rest of the text is too striking not to be addressed. It appears as if the Introduction was written before the results came in. Just to reiterate: this is careful, quantitative work, and is deserving of a more effective (and ideally shorter) delivery than is currently being given. Moreover, the readership of this journal (and other researchers working on related topics) will benefit from a straightforward description of the results, even if they seem less exciting that originally expected. In summary, directly saying "pulsar planets CANNOT form this way", is an informative result.

It was our historical context that carried us away. But we now rewrote the introduction to reflect the paper's content. And indeed, the introduction was probably written before we appreciated the final results.

We rewrote abstract and intro.

OTHER MAJOR COMMENTS:

  • Section 2.2, Second paragraph: Much like in Brandt & Podsiadlowski (1994), it would be useful to the reader to clarify that the semi-major axis a in the definition of E' is the old semi-major axis and distinct from the post- explosion semi-major axis a'.

Addressed

  • Section 2.2, Second paragraph: At the time of mass loss, why is the true anomaly nu_i chosen to be zero at all times? Should not the mean anomaly (i.e., time) be sampled randomly from 0 to 2pi? For eccentric binaries, choosing nu_0=0 is certainly not "without loss of generality" and might affect some of the results.

Clarified that this is only done in section 2 as part of coordinate choice, nu_i is randomly chosen in the simulation as can be seen in table 2.

Case in point: in Section 3.3, Sixth paragraph, the authors state that for the A,C pair to remain bound,"the remnant must receive a kick in the right direction and with the proper magnitude to either pass close-by the planet C, [...]". If a kick "in the right direction" is needed, how can this scenario not depend on the choice of nu_i at the time of mass loss and velocity kick?

Good point, this slipped beneath our radar. We have taken another look at our results and examined the effect of picking a random true anomly vs a random mean anomaly. We have recalculated our resulting probabilities and figure 5/6.

-Section 3.3, First paragraph Care to comment on how the Mardling+Aarset stability criterion compares to the Holman+Wiegert stability criterion (https://ui.adsabs.harvard.edu/abs/1999AJ....117..621H/abstract) which was especially derived for circumbinary systems?

The details are subtle and not realy relevant for our discussion here.

  • Section 3.3, Second paragraph:

The bullet point list of this section is very helpful to visualize the potential outcomes the authors are studying. However, the second category "dynamically unstable" could use further clarification.

Addressed.

According to the caption of figure 5 (the explanation should be in the main text as well), these are "[...]systems which remain fully bound but become dynamically unstable." If the Mardling & Aarseth triple stability criterion was used to flag these systems, then that citation should be made explicit when this category is introduced (the paper is cited elsewhere in the manuscript).

done.

Also, I would suggest careful wording here. If the post-explosion triple has total negative energy (i.e., remains bound on energetic grounds), and then becomes dynamically unstable with one object being kick out to infinity, there must be a left-over binary with negative energy. So, one cannot go from bound triple to fully dissolved triple from dynamical interactions alone (I am certain the authors know this basic fact, but the text is a bit ambiguous).

Addressed.

Finally, "and undergoes a three-body interaction" seems an unnecessary addition to "becomes dynamically unstable", especially since all triples, in a strict sense, are undergoing three-body interactions.

Addressed.

  • Section 3.3, Fifth paragraph:

The sentence: "[...] attribute the decrease in the fraction of (A,B) systems that stay bound for ao > 10^3 au to the systematically larger measure of ai in this region[...]" is difficult to parse. But it also difficult to see why it is worth pointing out why the purple curve decreases slightly for a_o > 1e3...

Removed from paper as it was not very relevant to start with.

  • Section 3.3, Eighth paragraph:

The authors state that the "probability for stable triples quickly drops for ao > 1e3 au [...]". Is this independent of the initial distribution of planetary semi-major axes? It is clear that distant planets, being weakly bound to begin with, should be more susceptible to being unbound when the central object loses mass. But what is not so clear is whether this 1000 au boundary scales with the initial outer tail of the planet semi-major axis distribution.

Removed from paper.

  • Section 3.3, Ninth paragraph: How are the authors defining a negative inclination? Relative inclinations are formally defined as arccos(n_i.n_o) where n_i, n_o are the normal vectors of the inner (binary) and outer (planet) orbits, respectively. Thus, relative inclination is always a positive quantity (0-90 is prograde and 90-180 is retrograde). This finding and the discussion around it are very confusing.

Inclination difference was originally taken as difference of inclinations compared to a common axis, changed this to absolute value of this difference.

-Section 4: This overall Section reads more like a Discussion than a Conclusion. What are the actual findings of this work? Paragraphs 1,2,3 contain raw findings, but they could be significantly condensed. Paragraph 4 and 5 are definitely Discussion material. Paragraph 6 is a conclusion. Paragraph 7 is mostly meandering caveats than bring down the momentum of the paper and end on a slightly negative tone (caveats are good, but ending with caveats is just weird).

Moved some material to a discussion section, and tried to simplify the remaining text.

-Section 4, Fourth paragraph: The authors skim over a very intriguing possibility (in my humble opinion): a second SN explosion in the triples that survived. In this case, the BC scenario could very well repeat itself and leave us with a pulsar planet. Even more intriguingly, the authors suggest that "the planet sticks to the inner compact-object binary until they merge due to the emission of gravitational waves". This scenario would circle back to some of my objections with the Introduction, i.e., the formation of pulsar planets. Alas, the authors conclude that the probability of this happening is small. Yet, I could not find where the numbers they use came from (except for 0.14).

Attempted to clarify where numbers came from. This scenario is certainly interesting, but relies on two things happening which each have a probability <10^(-2), so its probability is smaller than 10^(-4), which makes it unlikely to happen in our galaxy.

MINOR COMMENTS:

  • Section 1, First paragraph: If the authors do keep their initial discussion on pulsar planets, I recommend reading/referencing a fascinating early take on this systems by Phinney & Hansen 1993 (https://ui.adsabs.harvard.edu/abs/1993ASPC...36..371P/abstract) in which the survivabilty of planets beyond the main sequence is discussed.

We removed the initial discussion on planet pulsars.

  • Section 1, Second paragraph: I am well aware that the term "ionize a binary" is widely used in the binary evolution community (despite my own preference for simply "unbind"). However, the expression "ionizes the planet" can be easily misinterpreted. Please change to "ionizes the planetary orbit" and make sure that either "ionize" or "unbind" is consistently used throughout the paper for the sake of clarity: e.g., Figure 5 refers to "unbound" binaries. There is even a "dissociate" in Section 3.1

Should be uniformly 'bound/unbound' now.

  • Section 3.1 Second paragraph. What is the real purpose of training a smooth kernel on an empirical distribution? Bootstrap the number of systems? Benefit from the sample of a continuous (and differentiable) function? The benefit of this technique over simply resampling from an empirical histogram should be clearly stated, otherwise it seems like an unnecessary over-sophistication.

Bootstrap the number of systems. Changed this in the paper.

TYPOS:

The manuscript contains several typos and a few punctuation errors. Here are a few:

  • Section 1, second paragraph: might works -> might work

  • Section 1, third paragraph: larger then -> larger than

  • Section 3.2, first paragraph: zero binaries-> zero eccentricity binaries?

  • Section 3.3, last paragraph: neutral star -> neutron star

all fixed.

1- Streamline the paper. At the very least reword the Introduction and shorten /strengthen the done. Conclusions (or move part of the Conclusions to a Discussion section) done. 2 - Explain/fix the issue of negative inclinations done. 3- Justify/fix the choice of true anomaly nu_i=0 in their calculations done. 4 - Explain the category "unstable triple" better done.

---

## Round 2 · Referee Report · Anonymous (Referee 1) · 2021-12-5

Strengths

1 - This work is fairly novel: it carries out a largely unexplored idea of planet survivability around evolved, post-SN binaries.
2 - The work is careful and reliable: it combines known analytical calculations of orbital changes due to mass loss, with up-to-date population synthesis models.
3 - The updated manuscript is clear: despite some results that might be deemed "null" (rates inferred are low and perhaps observationally interesting), the text delivers the methods and results in a way that is clear and interesting to the reader.

Weaknesses

1 - Observational predictions are unclear, limiting the impact of this work and its ability to attract the interest of readers/observers. But this is partly due to observational uncertainties and thus beyond the scope of this papers. The authors do acknowledge in the abstract that their measured rates must be convolved with the planet bearing frequency of the high-mass star population (without actually carrying out that calculation, due to the largely unknown 2- The updated manuscript is too long. Despite the much improved language and pace of the introductory text, some of the descriptions of previous work are redundant and/or too detailed.

Report

Dear Scientific Editor:

I have carefully read the Scipost revised manuscript 2101.08033v2 entitled "Lucky planets: how circumbinary planets survive the supernova in one of the inner-binary components".

Based on SciPost Astronomy's acceptance criteria, this manuscript is publishable based on a broad interpretation of expectation #3: "Open a new pathway in an existing or a new research direction, with clear potential for multipronged follow-up work" Indeed --and despite the arguably modest findings of this work-- the authors have made strides in making quantitative predictions for (circumbinary) planets around evolved stars.

Moreover, after the text has been streamlined, I believe that the general criteria for publication are now satisfied. Namely; 1-"Be written in a clear and intelligible way..." The writing is now at a much more acceptable level 2 -"Contain a detailed abstract and introduction..." The abstract summarizes the results and goes straight to the point. The updated Introduction is much clearer than in the previous version. 3- "Provide sufficient details..." I was able to follow, understand or reproduce most of the quantitative steps of this paper. An expert would undoubtedly deem this paper fully reproducible. 4- "Provide citations to relevant literature" This is done in a satisfactory level. 5 - "Provide all reproducibility-enabling resources.." The use of the open-source code suite AMUSE ensure the reproducibility of the work 6- "Contain a clear conclusion summarizing the results and offering perspectives for future work." This requirement is satisfied.

---

## Round 2 · Author Response

It is a pleasure to respond to the referee report on our manuscript "Lucky planets: how circum-binary planets survive the supernova in one of the inner-binary components". about the survivability of planets in orbit around a binary in which one star experiences a supernova.

Here we respond to the referee's comments.

MAIN IMPRESSION: While certainly publishable, this paper could benefit from further streamlining its message: ideally, it should be direct and concise, and not weakened by its own counterpoints/caveats, or diluted by open ended speculations and pessimistic takes on how observable the theoretical predictions are.

We streamelined the paper, according to the referee's recommendation.

GENERAL COMMENTS: Heuristic or semi-quantitative arguments about "second generation" planets or "Phoenix" planets around stellar remnants have been proposed on-and-off for the past ~30 years. However, detailed Monte Carlo experiments are less numerous, and thus such degree of novelty makes this paper worthy of publication. However, the tone and purpose of this paper is ambiguous. The Introduction opens with a discussion on the pulsar planets, making the reader think that this paper is about mechanisms of producing such systems. If "pulsar planet production from pre-existing circumbinary planets" is indeed the hypothesis being tested here, then this paper is reporting a null result. And null results are fine. But neither in the abstract, the conclusion, nor anywhere in the narrative of the paper is categorically stated that the inferred hypothesis (perhaps I misunderstood the intended one) has been proven inadequate.

Thus, for good writing's sake, the authors need to decide if this is a null result (which is welcome) or something else, but the disconnect between the Introduction and the rest of the text is too striking not to be addressed. It appears as if the Introduction was written before the results came in. Just to reiterate: this is careful, quantitative work, and is deserving of a more effective (and ideally shorter) delivery than is currently being given. Moreover, the readership of this journal (and other researchers working on related topics) will benefit from a straightforward description of the results, even if they seem less exciting that originally expected. In summary, directly saying "pulsar planets CANNOT form this way", is an informative result.

It was our historical context that carried us away. But we now rewrote the introduction to reflect the paper's content. And indeed, the introduction was probably written before we appreciated the final results.

We rewrote abstract and intro.

OTHER MAJOR COMMENTS:

  • Section 2.2, Second paragraph: Much like in Brandt & Podsiadlowski (1994), it would be useful to the reader to clarify that the semi-major axis a in the definition of E' is the old semi-major axis and distinct from the post- explosion semi-major axis a'.

Addressed

  • Section 2.2, Second paragraph: At the time of mass loss, why is the true anomaly nu_i chosen to be zero at all times? Should not the mean anomaly (i.e., time) be sampled randomly from 0 to 2pi? For eccentric binaries, choosing nu_0=0 is certainly not "without loss of generality" and might affect some of the results.

Clarified that this is only done in section 2 as part of coordinate choice, nu_i is randomly chosen in the simulation as can be seen in table 2.

Case in point: in Section 3.3, Sixth paragraph, the authors state that for the A,C pair to remain bound,"the remnant must receive a kick in the right direction and with the proper magnitude to either pass close-by the planet C, [...]". If a kick "in the right direction" is needed, how can this scenario not depend on the choice of nu_i at the time of mass loss and velocity kick?

Good point, this slipped beneath our radar. We have taken another look at our results and examined the effect of picking a random true anomly vs a random mean anomaly. We have recalculated our resulting probabilities and figure 5/6.

-Section 3.3, First paragraph Care to comment on how the Mardling+Aarset stability criterion compares to the Holman+Wiegert stability criterion (https://ui.adsabs.harvard.edu/abs/1999AJ....117..621H/abstract) which was especially derived for circumbinary systems?

The details are subtle and not realy relevant for our discussion here.

  • Section 3.3, Second paragraph:

The bullet point list of this section is very helpful to visualize the potential outcomes the authors are studying. However, the second category "dynamically unstable" could use further clarification.

Addressed.

According to the caption of figure 5 (the explanation should be in the main text as well), these are "[...]systems which remain fully bound but become dynamically unstable." If the Mardling & Aarseth triple stability criterion was used to flag these systems, then that citation should be made explicit when this category is introduced (the paper is cited elsewhere in the manuscript).

done.

Also, I would suggest careful wording here. If the post-explosion triple has total negative energy (i.e., remains bound on energetic grounds), and then becomes dynamically unstable with one object being kick out to infinity, there must be a left-over binary with negative energy. So, one cannot go from bound triple to fully dissolved triple from dynamical interactions alone (I am certain the authors know this basic fact, but the text is a bit ambiguous).

Addressed.

Finally, "and undergoes a three-body interaction" seems an unnecessary addition to "becomes dynamically unstable", especially since all triples, in a strict sense, are undergoing three-body interactions.

Addressed.

  • Section 3.3, Fifth paragraph:

The sentence: "[...] attribute the decrease in the fraction of (A,B) systems that stay bound for ao > 10^3 au to the systematically larger measure of ai in this region[...]" is difficult to parse. But it also difficult to see why it is worth pointing out why the purple curve decreases slightly for a_o > 1e3...

Removed from paper as it was not very relevant to start with.

  • Section 3.3, Eighth paragraph:

The authors state that the "probability for stable triples quickly drops for ao > 1e3 au [...]". Is this independent of the initial distribution of planetary semi-major axes? It is clear that distant planets, being weakly bound to begin with, should be more susceptible to being unbound when the central object loses mass. But what is not so clear is whether this 1000 au boundary scales with the initial outer tail of the planet semi-major axis distribution.

Removed from paper.

  • Section 3.3, Ninth paragraph: How are the authors defining a negative inclination? Relative inclinations are formally defined as arccos(n_i.n_o) where n_i, n_o are the normal vectors of the inner (binary) and outer (planet) orbits, respectively. Thus, relative inclination is always a positive quantity (0-90 is prograde and 90-180 is retrograde). This finding and the discussion around it are very confusing.

Inclination difference was originally taken as difference of inclinations compared to a common axis, changed this to absolute value of this difference.

-Section 4: This overall Section reads more like a Discussion than a Conclusion. What are the actual findings of this work? Paragraphs 1,2,3 contain raw findings, but they could be significantly condensed. Paragraph 4 and 5 are definitely Discussion material. Paragraph 6 is a conclusion. Paragraph 7 is mostly meandering caveats than bring down the momentum of the paper and end on a slightly negative tone (caveats are good, but ending with caveats is just weird).

Moved some material to a discussion section, and tried to simplify the remaining text.

-Section 4, Fourth paragraph: The authors skim over a very intriguing possibility (in my humble opinion): a second SN explosion in the triples that survived. In this case, the BC scenario could very well repeat itself and leave us with a pulsar planet. Even more intriguingly, the authors suggest that "the planet sticks to the inner compact-object binary until they merge due to the emission of gravitational waves". This scenario would circle back to some of my objections with the Introduction, i.e., the formation of pulsar planets. Alas, the authors conclude that the probability of this happening is small. Yet, I could not find where the numbers they use came from (except for 0.14).

Attempted to clarify where numbers came from. This scenario is certainly interesting, but relies on two things happening which each have a probability <10^(-2), so its probability is smaller than 10^(-4), which makes it unlikely to happen in our galaxy.

MINOR COMMENTS:

  • Section 1, First paragraph: If the authors do keep their initial discussion on pulsar planets, I recommend reading/referencing a fascinating early take on this systems by Phinney & Hansen 1993 (https://ui.adsabs.harvard.edu/abs/1993ASPC...36..371P/abstract) in which the survivabilty of planets beyond the main sequence is discussed.

We removed the initial discussion on planet pulsars.

  • Section 1, Second paragraph: I am well aware that the term "ionize a binary" is widely used in the binary evolution community (despite my own preference for simply "unbind"). However, the expression "ionizes the planet" can be easily misinterpreted. Please change to "ionizes the planetary orbit" and make sure that either "ionize" or "unbind" is consistently used throughout the paper for the sake of clarity: e.g., Figure 5 refers to "unbound" binaries. There is even a "dissociate" in Section 3.1

Should be uniformly 'bound/unbound' now.

  • Section 3.1 Second paragraph. What is the real purpose of training a smooth kernel on an empirical distribution? Bootstrap the number of systems? Benefit from the sample of a continuous (and differentiable) function? The benefit of this technique over simply resampling from an empirical histogram should be clearly stated, otherwise it seems like an unnecessary over-sophistication.

Bootstrap the number of systems. Changed this in the paper.

TYPOS:

The manuscript contains several typos and a few punctuation errors. Here are a few:

  • Section 1, second paragraph: might works -> might work

  • Section 1, third paragraph: larger then -> larger than

  • Section 3.2, first paragraph: zero binaries-> zero eccentricity binaries?

  • Section 3.3, last paragraph: neutral star -> neutron star

all fixed.

1- Streamline the paper. At the very least reword the Introduction and shorten /strengthen the done. Conclusions (or move part of the Conclusions to a Discussion section) done. 2 - Explain/fix the issue of negative inclinations done. 3- Justify/fix the choice of true anomaly nu_i=0 in their calculations done. 4 - Explain the category "unstable triple" better done.

---

## Round 2 · List of Changes

See above

---

## Editorial Decision

published